# Macrocyclic Diterpenoids from Euphorbiaceae as A Source of Potent and Selective Inhibitors of Chikungunya Virus Replication

**DOI:** 10.3390/molecules24122336

**Published:** 2019-06-25

**Authors:** Simon Remy, Marc Litaudon

**Affiliations:** Institut de Chimie des Substances Naturelles, CNRS ICSN, UPR 2301, Université Paris Saclay, 91198 Gif-sur-Yvette, France; simon.remy@cnrs.fr

**Keywords:** chikungunya, Euphorbiaceae, phorbol, tigliane, daphnane, ingenane, jatrophane, pre-myrsinane, flexibilane, PKC

## Abstract

Macrocyclic diterpenoids produced by plants of the Euphorbiaceae family are of considerable interest due to their high structural diversity; and their therapeutically relevant biological properties. Over the last decade many studies have reported the ability of macrocyclic diterpenoids to inhibit in cellulo the cytopathic effect induced by the chikungunya virus. This review; which covers the years 2011 to 2019; lists all macrocyclic diterpenoids that have been evaluated for their ability to inhibit viral replication. The structure–activity relationships and the probable involvement of protein kinase C in their mechanism of action are also detailed.

## 1. Introduction

Chikungunya virus (CHIKV) is an arthropod-borne virus causing an infectious disease characterized by fever, arthralgia and, sometimes, a maculopapular rash [1]. Despite high morbidity rate, there is currently no approved vaccine or antiviral treatment. Different classes of compounds that target either a viral or a host factor have been reported to inhibit CHIKV replication in vitro [2], but none have progressed to further development. The development of potent antiviral drugs against CHIKV is, therefore, urgently needed.

Plant species of the Euphorbiaceae family are known to produce a vast array of macrocyclic diterpenoids. They possess various types of carbon skeletons (e.g., jatrophane, lathyrane, myrsinane, ingenane, tigliane, daphnane etc.), many of which are unique to this large plant family [3,4]. The genus *Euphorbia*, which includes more than 2000 species, alone provided several hundred macrocyclic diterpenoids, representative of more than 20 skeletal types [5,6]. With regard to their biogenesis, these compounds also called lower diterpenoids are formed by a head-to-tail cyclization of a tetraprenylphosphate precursor through the agency of a type I cyclase, leading to the formation in a step-wise fashion of the monocyclic cembranes, the bicyclic casbanes and the tricyclic lathyrane derivatives, the latter may lead to jatrophane after cyclopropane ring opening [7]. From a structural standpoint phorboids, i.e., tiglianes, daphnanes, ingenanes and rhamnopholanes, are polycyclic diterpenoids but are biogenetically derived from macrocyclic precursors. However, Appendino pointed out that the relationship between macrocyclic diterpenoids and phorboids has remained a mechanistic black box, despite obvious structural similarities between the lathyrane and the tigliane skeletons [7].

Macrocyclic diterpenoids are of considerable interest due to their therapeutically relevant biological properties, one of the most widely described being their ability to modulate protein kinase C (PKCs) activities [4,5]. PKCs are involved in many physiological functions [8]. In addition, it has been shown that biologically active diterpenes containing a *gem*-dimethylcyclopropane subunit such as lathyrane-, casbane-, premyrsinane-, ingenane- or tigliane-type diterpenoids are an intriguing source of PKC modulators [9]. PKC activation is responsible for a wide variety of biological activities such as platelet aggregation, tumor-promotion and anti-HIV activity.

Since the discovery of their anti-HIV properties in the early 1990s, macrocyclic diterpenoids have attracted the interest of the scientific community [10,11,12,13,14] and a significant number of studies have focused on their anti-HIV potential through PKC activation [15,16,17,18]. Some non-tumor-promoting tiglianes such as prostratin and DDP (12-deoxyphorbol-13-phenylacetate) exhibit potent in vitro activity toward the induction of HIV expression in latently infected cell lines and primary cells [19] and are considered to be promising anti-HIV agents [20].

In contrast, the discovery of antiviral properties of macrocyclic diterpenes against CHIKV is much more recent [20,21,22,23,24] as it followed the chikungunya African outbreak in the 2000s and its subsequent emergence in the Indian Ocean [24]. The main vectors are *Aedes* mosquitoes from the Culicidae family such as *A. furcifer*, *A. africanus*, *A. luteocephalus*, *A. taylori* and *A. aegypti*. Between 2005 and 2006, a new vector (*A. albopictus*) spread in most of the tropical and subtropical areas and led to massive outbreak in the Indian Ocean region.

As CHIKV reached epidemic level, the quest for novel and selective antiviral compounds was launched on a large scale. A project entitled ‘Biodiversity and emerging viruses in the Indian Ocean: selection of drug candidates targeting the Chikungunya virus′, was selected by the Centre for Research and Monitoring of Emerging Diseases in the Indian Ocean (CRVOI) for financial support and was developed between 2009 and 2011 [24]. The main objective was to discover and characterize new selective antiviral compounds from the plant biodiversity of the Indian Ocean (Madagascar, Reunion and Mauritius), which was then extended to that of New Caledonian and Mediterranean flora. The research program, led by a tight network of virologists and natural products chemists, quickly revealed the Euphorbiaceae as the most promising plant family in the fight against CHIKV.

This review focuses on the anti-CHIKV activity of about 80 naturally occurring macrocyclic diterpenoids isolated from plant species belonging to the Euphorbiaceae family from 2011 to 2019, along with about 30 commercially available natural diterpenoids. These compounds, which have been classified according to their chemical features into 14 skeletal types have all been evaluated using the methodology described by Bourjot et al. [21]. The discussion focuses on the structure–activity relationships that are detailed when a sufficient number of compounds have been tested in each series. The mechanism of action of the most promising compounds that involve PKCs is also discussed, highlighting the close analogy with their anti-human immunodeficiency virus (HIV) activities.

## 2. Tiglianes and Ingenanes

Tigliane diterpenoids form the largest group of phorboids. They possess a 5/7/6/3-tetracyclic ring system, in which rings A and B, and B and C, are *trans*-fused while rings C and D are *cis*-fused. A carbonyl is located at C-3, a double-bond at C-1, and most of the tiglianes are hydroxylated in positions 4, 9, 12, 13, and 20 [5]. Also called phorbol esters (PE), most tigliane derivatives exist in the form of 12,13 or 13,20-diesters, and a few also exist as 12- or 13-monoesters and 12,13,20-triesters. They are classified as (i) phorbol esters including 12- and 13-monoesters, 12,13- and 13,20-diesters, and 12,13,20-triesters, (ii) 4-deoxyphorbol esters, (iii) 12-deoxyphorbol esters, (iv) 4,12- dideoxyphorbol esters and (v) 4,20-dideoxyphorbol esters [4]. Ingenane diterpenoids, which are a biogenetically advanced group of phorboids [7], possesses a scaffold composed of a 5/7/7/3-tetracyclic ring system including a ketone bridge between C-8 and C-10 and are β-hydroxylated at C-4. Rings A and B are *trans*-fused and double bonds can be found between C-1 and C-2 in ring A, and between C-6 and C-7 in ring B. The C-3, C-5, C-13, C-17, and C-20 positions can be oxygenated and/or esterified [5].

A total of 51 tiglianes (**1**–**51**) and three ingenanes (**52**–**54**) has been evaluated in a virus-cell-based assay against CHIKV (Figure 1). These compounds were either commercially available.

(**1**–**20**,**25**,**26**,**41**–**43**,**52**–**54**) [16] or isolated from various Euphorbiaceae species i.e., *Trigonostemon howii* (**21**) [21], *Croton mauritianus* (**23**,**24**) [25], *Euphorbia semiperfoliata* (**33**–**35**,**40**) [26], *Stillingia lineata* (**22**,**31**,**32**,**44**,**45**) [27], *Bocquillonia nervosa* (**46**–**49**) [28], *Euphorbia pithyusa* (**42**,**43,50,51**) [29], *Euphorbia dendroides* (**36**–**39**) [30], *Euphorbia cupanii* (**27**–**30**) [31]. Additionally, phorbols have also been identified in *Sandwithia guyanensis* [32] and *Sagotia racemosa* bark extracts as potent putative anti-CHIKV agents.

First, the cytotoxicity of all compounds was evaluated against African green monkey kidney epithelial cell line (Vero cells). The CC50 (50% antimetabolic concentration) values ranged from 4.1 to > 343 µM, phorbol (**1**) being the less cytotoxic compound. Among compounds with a selective index > 20 (see below), the highest cytotoxicity was obtained for compounds with a long acyl chain either at C-12 or C-13 position (**11**, **15** and **48**).

Most diterpenes have shown significant CHIKV inhibitory activities but the level of activity seems to be highly dependent on the structural type and its decoration (Table 1). Phorbol-12,13-didecanoate (**11**), 12-*O*-tetradecanoylphorbol-13-acetate (TPA, **15**) and to a lesser extent 12-deoxyphorbol-13-hexadecanoate (**46**) were found to be the most potent inhibitors yet reported as evidenced by their lower EC50 (effective concentration or concentration which is calculated to inhibit virus induced cell death by 50%) and higher selectivity indices values (EC50 = 6.0, 2.9 and 20 nM, and SI = 686, 1965 and 1500, respectively). Interestingly TPA did not show any significant antiviral activities against Sindbis virus (SINV) and Semliki Forest virus (SFV), two other members of the genus *Alphavirus* [21]. Thirty-two other tiglianes and one ingenane have shown significant anti-CHIKV activities with EC50 values between 20 nM and 5 µM. They belong to all structural sub-classes defined previously. Among these, phorbol esters **22**, and **27**–**29**, 4-deoxyphorbol esters **33**, **35**, **37** and **38**, and 12-deoxyphorbol esters **41** (prostratin), **44**, and **47**–**49** exhibited selective indices >20.

Although it is difficult to draw clear structure–activity relationships within this family of compounds, general rules can be highlighted. As was previously reported [16,26], the anti-CHIKV activity of some phorbols can be modulated by the length and location at C-12 and/or C-13 positions of the acyl chain(s) on the phorbol backbone, the relative configuration at C-4 and the presence of additional carbonyl function at C-7 and/or C-20 (Figure 2). Generally, a stronger anti-CHIKV activity was reported for phorbol monoesters (**3**,**6**,**7**) and phorbol di- and triesters (**9**,**11**,**15**,**17**, **19**, and **26**–**30**) possessing long aliphatic side chains at C-12 and/or C-13. Conversely, those having short side chains at C-12 and C-13 are less active and less selective, with the exception of compound **22** possessing an acetyl side chain and a 2-methylbutyryl side chain at C-12 and C-13, respectively (EC50 = 3.3 µM, and SI = 41). Comparison of the anti-CHIKV activities of compounds **11** and **12**, and **15** (TPA) and **16** indicated that 4β-phorbol derivatives are much more potent than their 4α-counterparts. A similar observation can be noted by comparing the antiviral activities of the 4α-deoxyphorbols **31** and **34**, much weaker than those of 4β-deoxyphorbols **35** and **33**, all possessing short side chains in C-12 and C-13. The 4α-deoxyphorbols **31** and **34** showed lower activities. In contrast, 4α-deoxyphorbol **32**, which possess a nona-2-enoyl side chain at C-13, showed a significant anti-CHIKV activity (EC50 = 1.4 µM, and SI = 5.1). It should be noted that compound **32** as well as compound **45** also showed a significant antiviral activity on the replication of SINV [27]. All 4β-deoxyphorbols but compound **39**, exhibited potent anti-CHIKV activity, among which 4β-deoxyphorbol 12-acetate-13-isobutyrate (**35**) has the highest selective index (EC50 = 0.44 µM, and SI = 390). Finally, most of the 12-deoxyphorbols have shown strong anti-CHIKV activities. In particular, compounds **46**–**48**, which all have an hexadecanoyl side chain at C-13, exhibited the best indices of selectivity. Among the latter, the presence of a 6,7-epoxy function instead of a 6,7-dihydroxy moiety has a favorable effect on the anti-CHIKV activity (**48** vs. **49**). [28] In general, oxidation of phorbol esters [33] leading to an α,β-unsaturated carbonyl function at C-20 or C-7 appears to be responsible for a significant decrease in antiviral activity (**25** vs. **9**, **26** vs. **15**, **40**, **21**). Finally, it should be noted that ingenol-3,20-dibenzoate **54**, was the first ingenane-type diterpenoid showing a significant anti-CHIKV activity [16].

Most of tigliane and ingenane derivatives mentioned in Table 1 have also been evaluated against HIV-1 and HIV-2 replications [16,27,28]. Overall, the structure–activity relationships that were established for the anti-CHIKV activities of the tested phorbol derivatives were found to be similar to those observed for anti-HIV-1 and anti-HIV-2. This concerns the role of the length and the position of the acyl chains at C-12 and C-13, the requirement of a C-4β configuration for a strong antiviral effect, and the deleterious effect of an oxidation at C-20. In particular the close antiviral profiles of the tested compounds against CHIKV on one hand and HIV-1 and HIV-2 on the other hand have been confirmed by the calculation of Pearson correlation coefficients between the EC50 values for each virus pair. The results showed that EC50s against CHIKV and HIVs were positively correlated (CHIKV/HIV-1, r = 0.81 ± 0.09; CHIKV/HIV-2, r = 0.84 ± 0.07) [16]. The authors concluded that, similarly to the mechanism of action proposed for HIV inhibitors, phorbol ester derivatives could operate according to a CHIKV-specific mechanism possibly associated with the activation of PKCs [34,35].

## 3. Daphnanes

Daphnane diterpenoids, which are believed to be derived from a tigliane precursor [36] are based on a 5/7/6-tricyclic skeleton, rings A and B and rings B and C being *trans*-fused. A double bond or an epoxy group may be present between C-6 and C-7 carbons. A large number of daphnane diterpenoids possess an orthoester moiety (Daphnane Diterpenoid Orthoesters, DDO), which can be attached on ring C at various positions i.e., C-9, C-11, C-12, C-13, and C-14 [36,37].

In several recent studies, the anti-CHIKV activities of the commercially available resiniferatoxin (**55**) [16], as well as DDOs isolated from *Trigonostemon cherrieri* (**56**-**63**) [22,23,38], *Neoguillauminia Cleopatra* (**64**) [28], and *Codiaeum peltatum* (**65**,**66**) [37], have been reported (Figure 3). Most of them have shown significant anti-CHIKV activities with EC50 values ranging from 0.6 to 18 µM (Table 2). From this chemical series, trigocherrierin A (**56**) possessing a 2-methyl-decanoyl side chain at C-12, and a 9,13,14-orthoester moiety exhibited the strongest antiviral activity with the highest selective index (EC50 = 0.6 µM, and SI = 72). Finally, it has been shown that anti-HIV activities of trigocherriolides are 100 to 1000 times higher than those of trigocherrins, suggesting a different mechanism of action [39]. Interestingly, compounds **59**–**62**, and to a lesser extent compounds **57** and **58**, showed significant antiviral activities on the replication of SINV and SFV viruses. [23] Finally, compounds **57**, **60** and **61** also showed significant inhibitory activity against NS5 RNA-dependent RNA polymerase of dengue virus (DENV) [23].

## 4. Jatrophanes

Jatrophane diterpenes are based on a 5/12-bicyclic ring system. The number of substitutable positions on the bicyclic core, provides jatrophanes with a great chemical diversity [5].

In 2014 and 2016, 25 jatrophanes (Figure 4) were isolated from *Euphorbia amygdaloides* ssp. *semiperfoliata* [40], and *Euphorbia dendroides* [41]. Their anti-CHIKV activities are reported in Table 3. Within the 9,14-dioxojatropha-dienes (**67**–**73**), an acetyl group at position 2 proved to be deleterious for anti-CHIKV activity (**69** vs. **72**, and **70** vs. **73**). Regarding compounds **67**–**70** and **71**–**75**, the authors ranged the influence of the C-8 substitution on the activity as follows: tiglyloxy > benzoyloxy > acetyloxy ≈ isobutyryloxy. In the 9-oxojatropha-dienes series, the 2-methylbutyryl group of **76** seemed to be deleterious for the antiviral activity (**76** vs. **74** and **75**). It should be noted that compound **69** exhibited moderate anti-SINV activity, while compounds **74**–**76** exhibited significant, albeit weak, antiviral activities on the replication of SINV and SFV viruses [40].

## 5. Myrsinanes and Premyrsinanes

Myrsinane and premyrsinane diterpenes possess a 5/7/6-fused tricyclic or a 5/7/6/3-fused tetracyclic skeleton, respectively. Rings A and B and rings B and C are *trans*-fused in both series, and an additional cyclopropane ring is present in premyrsinanes. Myrsinanes generally possess ester groups at positions C-3, C-5, C-7, and C-15, and a double bond between C-8 and C-9. In premyrsinanes, an hemiacetal ring or a 13/17-epoxy function can be present [5]. Seven premyrsinols (**93**–**99**) and one myrsinol (**100**) (Figure 5) were evaluated on the CHIKV-cell-based assay (Table 4). None of the compounds tested except compound **98** (EC50 = 11 µM, SI = 5.8) showed significant anti-viral activity. In the premyrsinol series, all compounds but **98** possess an ester group at C-7. It was suggested that this ester group could have a deleterious effect on anti-CHIKV activity [29].

## 6. Flexibilanes

Flexibilanes are rare 15-membered macrocyclic diterpenes that possess an intramolecular furan, a hydroxy group in position 10 and a side chain in position 8. In general, a pyranol ring complement their structural features. The structure of flexibilanes is rigid due to strong internal hydrogen bonds between the hydroxy group at C-10 and the ester oxygens of the side chain. The presence of five methyl groups on the macrocycle increases the rigidity of the flexibilanes [42,43]. Among the 10 flexibilanes evaluated (**101**–**110**) (Figure 6), tonantzitlolones B, C and F (**102, 103** and **106**) showed moderate anti-CHIKV activities with EC50 values of 12, 24 and 19 µM, and SI = 10.2, >9 and 3, respectively (Table 5) [27]. Compound **106** also showed moderate anti-SINV and anti-SFV activities.

## 7. Protein Kinase C (PKCs) as Targets of Phorbol Esters for Inhibition of Chikungunya Virus (CHIKV)

Recently, it has been shown that HIV-1 and HIV-2 inhibitory effects of phorbols esters were strongly correlated with those occurring on CHIKV [16]. These results were quite surprising given the fact that CHIKV and HIV belong to two different virus genera, *Alphavirus* and *Lentivirus,* respectively, but most probably can be explained through a common PKC-based mechanism of action. Although the mechanism remains poorly defined, this provides evidences that inhibition of CHIKV-induced cell death of phorbol esters might result from an activation of PKCs [44], and that PKC is an important target in CHIKV replication.

Protein kinase C (PKC) is a family of related serine/threonine kinases that regulate many cellular processes such as proliferation, differentiation and apoptosis. They have been classified into several distinct subfamilies depending on their specific requirements for activation. Classical isoforms (α, βI, βII, and γ) require calcium and diacylglycerol (DAG); novel isoforms (PKC-δ, -ε, -η, and -θ) require DAG but not calcium for activation, while activation of the atypical isoforms (Mζ- ι/λ isoforms) is independent of calcium and DAG. Following activation, PKCs undergo translocation from the cytoplasm to the plasma membrane and act trough phosphorylation of downstream signaling factors [45,46,47]. Due to their structural similarity with DAG, phorbol esters are powerful ligands of the regulatory domain of all classical and novel PKC isoforms.

The interaction of phorbols with PKC is dependent on their substitution pattern and requires a combination of optimal hydrogen bonding and hydrophobic contacts for high potency. Phorbols bind to a cysteine-rich site replacing a molecule of water and establishing hydrogen bond interactions through the oxygen atoms bound to carbons C-3, C-4, and C-20 [48,49,50]. The hydrophobic acyl chains of phorbol esters allow complex formation with PKCs and their anchoring to the membrane [50]. Changes on the C-3 oxygen atom led to lower PKC activation due to the loss of inductive and steric effects exerted on the C-4 hydroxy group [49,51]. Since the *cis*-configuration of the A/B rings junction might create a spatial arrangement incompatible with PKCs binding, the most potent PEs that modulate PKCs activity belong to the β-series (*trans*-fused A/B rings) [52]. By using computer-assisted modeling, it has been shown that the pharmacophore model for PEs required a hydrophobic region consisting of acyl substituents on C-12 and/or C-13 and a cyclohexane- cyclopropane-annellated ring system, and a hydrophilic domain spanning the C-3 to C-9 region including four groups able to form hydrogen bonds. An adequate distance and orientation of the ring system relative to membrane lipid bilayer are also necessary [51,53].

Tigliane diterpenoids, are one of the most important classes of diterpenoids from the Euphorbiaceae family [4,5]. Among tiglianes, prostratin is a 12-deoxyphorbol ester that has been demonstrated to be a potent activator of PKCs [47]. This compound, which has no pro-tumoral effect [10], was reported to inhibit the entry of HIV and to compromise latent HIV viral reservoirs through PKC-dependent mechanisms [54,55]. During the past 20 years many studies showed that phorbol derivatives stimulate HIV replication while inhibiting virion formation thus suppressing viral latent reservoirs through the same “kick and kill” strategy [13,34,35,56,57,58]. Recently, prostratin was shown to be a potent and selective inhibitor of CHIKV [21,44]. In particular, Neyts and colleagues demonstrated that its antiviral activity was dependent on the multiplicity of infection of the virus, and proved to be strongly dependent on the cell type [44]. A potent antiviral activity was observed in human skin fibroblast cells, the primary target cells of CHIKV infection. Prostratin mainly inhibits CHIKV replication at the post-entry stage hence exhibit antiviral activity when added to cells several hours post-infection. When tested in association with PKC inhibitors of known spectrum, the effect of prostratin appeared to be mediated mainly by the activation of classical PKC isoforms [44].

Considering the anti-CHIKV activity of PEs related to the present work, the structure–activity relationships have suggested the importance of the C-4 configuration, the influence of carbonyl at C-20 or C-7 and 6,7-epoxy function (which is spatially close to C-20) and the key role of acyl chains (See Section 1). These assumptions are in complete agreement with the pharmacophore model developed for phorbol-PKCs interactions [44,45,46,47,48].

A major concern about phorbols is their pro-tumoral effect. It has been attributed to hydrophobic acyl chains that exposed outward from the PKC/phorbol complex and likely retain complexes at the plasma membrane. This results in a sustained PKC activation which ultimately lead to the loss of its regulatory activity. Among PEs, a long side chain in position 13 associated with the absence of hydroxyl in position 12 (12-deoxyphorbol esters) seemed to be responsible for a strong tumor-promoting activity. Accordingly, PEs bearing short or medium acyl chain(s) may activate classical PKCs and should be devoid of tumor promoter activity. This sub-class of compounds might ideally be considered for anti-CHIKV compounds development [15,35,59,60].

## 8. Conclusions

Macrocyclic diterpenoids are an important source of lead compounds for drug development [61,62,63]. The non-tumor-promoting tiglianes, prostratin and DDP, are considered to be promising anti-HIV agents [20]. Ingenol mebutate a natural product identified from *Euphorbia peplus* is used as a topical gel (Picato^®^) for treatment of keratose actinic [61,62,63]. Others are currently in preclinical or clinical studies. Tigilanol tiglate (EBC-46^®^) has completed safety and efficacy studies for the treatment of solid tumors in dogs [64] and is currently in clinical study for the treatment of head and neck tumors in human adults [65].

In this review, we have shown than macrocyclic diterpenoids can also provide compounds with powerful antiviral activities. In particular, phorbol esters, 4-deoxy and 12-deoxyphorbol esters proved to be among the most promising anti-CHIKV agents yet reported. Considering that PKCs are potential host targets for the inhibition of CHIKV replication and that PEs most likely act though a PKC-dependent pathway, these compounds and their analogs offer interesting development opportunities as potential therapeutic agents for chikungunya treatment.

## Figures and Tables

**Figure 1 molecules-24-02336-f001:**
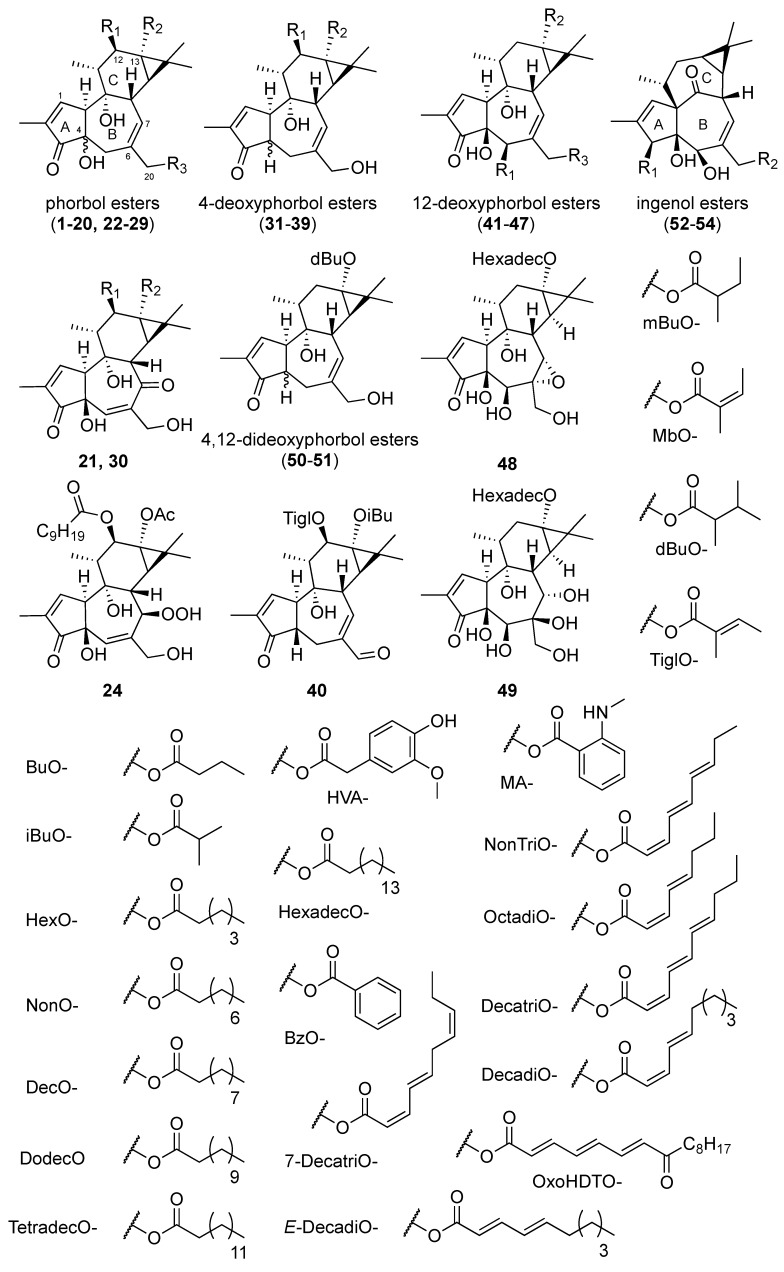
Structures of tiglianes **1**–**51** and ingenanes **52**–**54**.

**Figure 2 molecules-24-02336-f002:**
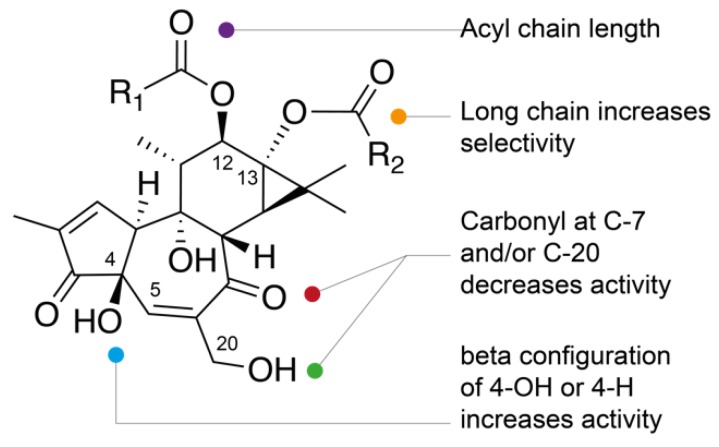
Structure–activity relationships of tiglianes.

**Figure 3 molecules-24-02336-f003:**
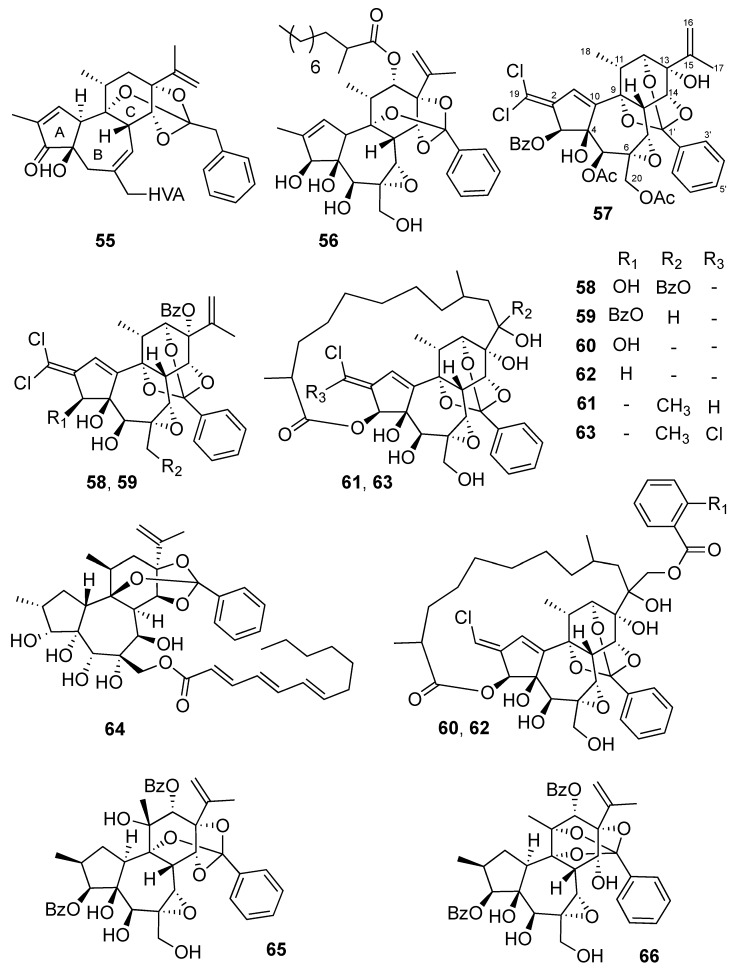
Structures of daphanes **55**–**66**.

**Figure 4 molecules-24-02336-f004:**
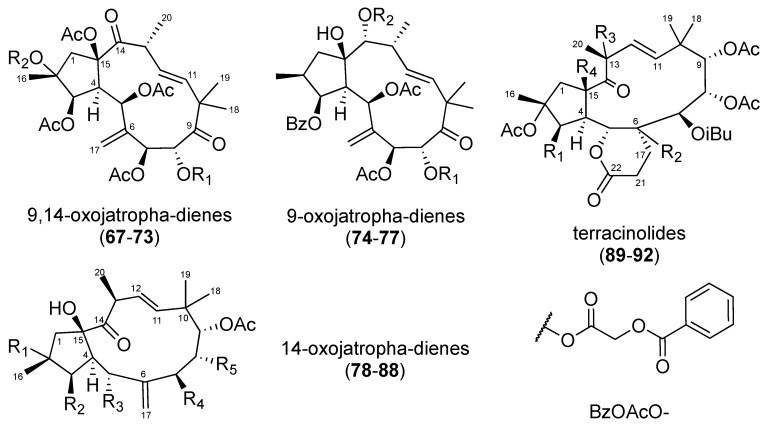
Structures of jatrophanes **67**–**92**.

**Figure 5 molecules-24-02336-f005:**
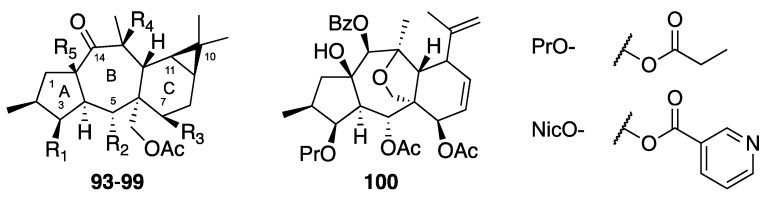
Structures of premysinanes **93**–**99** and myrsinane **100**.

**Figure 6 molecules-24-02336-f006:**
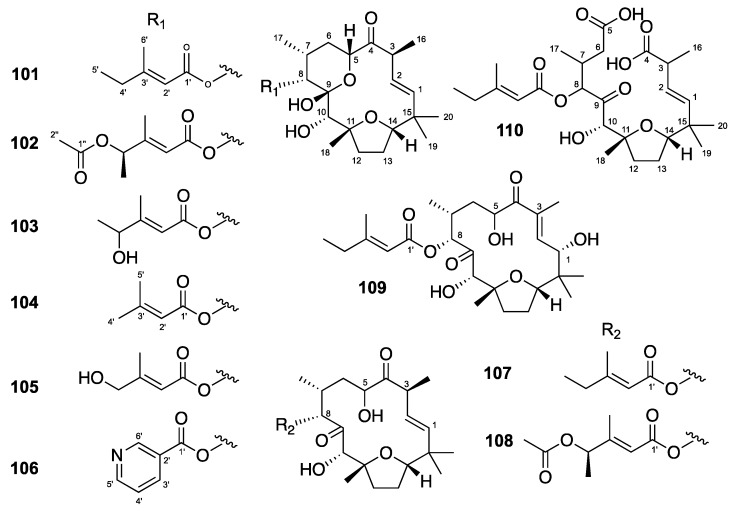
Structures of flexibilanes **101**–**110**.

**Table 1 molecules-24-02336-t001:** Anti-chikungunya virus (CHIKV) activities of tiglianes **1**–**51** and ingenanes **52**–**54**.

	Compound	CHIKV EC50	SI	R1	R2	R3
	**Phorbol esters**					
**1**	Phorbol	>343	>1	HO-	HO-	HO-
**2**	Phorbol-12-acetate	>245	0.8	AcO-	HO-	HO-
**3**	Phorbol-12-decanoate	4.9 ± 1.7	1.5	DecO-	HO-	HO-
**4**	Phorbol-13-acetate	>174	n.d.	HO-	AcO-	HO-
**5**	Phorbol-13-butyrate	20 ± 10	12.3	HO-	BuO-	HO-
**6**	Phorbol-13-decanoate	2.2 ± 0.1	9.7	HO-	DecO-	HO-
**7**	Phorbol-13-tetradecanoate	0.99 ± 0.03	9.0	HO-	TetradecO-	HO-
**8**	Phorbol-12,13-diacetate	9.4 ± 1.0	4.2	AcO-	AcO-	HO-
**9**	Phorbol-12,13-dibutyrate	1.8 ± 0.2	2.1	BuO-	BuO-	HO-
**10**	Phorbol-12,13-dihexanoate	3.2 ± 0.2	1.8	HexO-	HexO-	HO-
**11**	Phorbol-12,13-didecanoate	6.0 ± 0.9 nM	686	DecO-	DecO-	HO-
**12**	4α-Phorbol-12,13-didecanoate	1.5 ± 0.1	2.2	DecO-	DecO-	HO-
**13**	Phorbol-13,20-diacetate	24.6 ± 7.1	1.7	HO-	AcO-	AcO-
**14**	Phorbol-12,13,20-triacetate	32.6 ± 4.0	2.2	AcO-	AcO-	AcO-
**15**	12-*O*-Tetradecanoylphorbol-13-acetate (TPA)	2.9 ± 0.3 nM	1965	TetradecO-	AcO-	HO-
**16**	12-*O*-Tetradecanoyl-4α-phorbol-13-acetate	2.8 ± 0.5	1.9	TetradecO-	AcO-	HO-
**17**	12-*O*-Tiglylphorbol-13-decanoate	1.1 ± 0.3	3.3	TiglO-	DecO-	HO-
**18**	12-*O*-(*N*-methylanthranilate)-phorbol-13-acetate	15 ± 4	1.1	MA-	AcO-	HO-
**19**	12,13-*O,O′*-Dinonanoylphorbol-20-homovanillate	0.6 ± 0.1	3.7	NonO-	NonO-	HVA-
**20**	12-*O*-Phenylacetyl-13-*O*-acetylphorbol-20-homovanillate	1.7 ± 0.3	14.2	PhAcO-	AcO-	HVA-
**21**	Trigowiin A	>100	>2.3	DodecO-	AcO-	-
**22**	12-*O*-Acetylphorbol-13(2″-methyl)-butyrate	3.3 ± 0.3	41	AcO-	mBuO-	HO-
**23**	12-*O*-Decanoylphorbol-13-acetate	2.4 ± 0.3	2.0	DecO-	AcO-	HO-
**24**	12-*O*-Decanoyl-7-hydroperoxy-5-ene-13-acetate phorbol	4.0 ± 0.8	1.9	-	-	-
**25**	20-Oxo-phorbol-12,13-dibutyrate	13.1 ± 0.5	2.7	BuO-	BuO-	O=
**26**	20-Oxo-TPA	0.7 ± 0.1	5.9	TetradecO-	AcO-	O=
**27**	12β-*O*-[Deca-2*E*,4*Z*-dienoyl]-13α-isobutyl-4β-phorbol	<0.7	>77	*E*-DecadiO-	iBuO-	HO-
**28**	12β-*O*-[Deca-2*E*,4*Z*-dienoyl]-13α-(2-methylbutyl)-4β-phorbol	<0.7	>12	*E*-DecadiO-	mBuO-	HO-
**29**	12β-*O*-[Deca-2*Z*,4*E*-dienoyl]-13α-isobutyryl-4β-phorbol	<0.8	>58	DecadiO-	iBuO-	HO-
**30**	12β-*O*-[Deca-2*Z*,4*E*-dienoyl]-13α-isobutyryl-5-ene-7-oxo-4β-phorbol	4.5 ± 0.6	6	DecadiO-	iBuO-	-
	**4-Deoxyphorbol esters**					
**31**	12β-*O*-Acetyl-4α-deoxyphorbol-13(2″-methyl)-butyrate	77	1.4	AcO-	mBuO-	-
**32**	12β-*O*-[Nona-2*Z*,4*E*,6*E*-trienoyl]-4α-deoxyphorbol-13-butyrate	1.4 ± 0.2	5.1	NontriO-	BuO-	-
**33**	4β-Deoxyphorbol-12-tiglate-13-isobutyrate	1.0 ± 0.4	25	TiglO-	iBuO-	-
**34**	4α-Deoxyphorbol-12-tiglate-13-isobutyrate	17.0 ± 1.0	7	TiglO-	iBuO-	-
**35**	4β-Deoxyphorbol-12-acetate-13-isobutyrate	0.44 ± 0.03	390	AcO-	iBuO-	-
**36**	12β-*O*-[Deca-2*Z*,4*E*-dienoyl]-13α-isobutyryl-4β-deoxyphorbol	0.9 ± 0.1	6	DecadiO-	iBuO-	-
**37**	12β-*O*-[Deca-2*Z*,4*E*,6*E*-trienoyl]-13α-isobutyryl-4β-deoxyphorbol	0.6 ± 0.6	41	DecatriO-	iBuO-	-
**38**	12β-*O*-[Octa-2*Z*,4*E*-dienoyl]-13α-isobutyryl-4β-deoxyphorbol	0.4 ± 0.02	34	OctaDiO-	iBuO-	-
**39**	12β-*O*-[Deca-2*Z*,4*E*,7*Z*-trienoyl]-13α-isobutyryl-4β-deoxyphorbol	12.6 ± 46.2	4	7-DecatriO-	iBuO-	-
	**4,20-Dideoxyphorbol ester**					
**40**	4α,20-Dideoxyphorbol-12-tiglate-13-isobutyrate	51.1 ± 4.1	3	-	-	-
	**12-Deoxyphorbol esters**					
**41**	12-Deoxyphorbol-13-acetate (prostratin)	2.7 ± 1.2	22.8	H-	AcO-	HO-
**42**	13-*O*-Isobutyryl-12-deoxyphorbol-20-acetate	0.7 ± 0.1	5.0	H-	BuO-	AcO-
**43**	13-*O*-Phenylacetyl-12-deoxyphorbol-20-acetate	50.8 ± 2.1	1.9	H-	PhAcO-	AcO-
**44**	12-Deoxyphorbol-13(2″-methyl)butyrate	1.2 ± 0.2	>240	H-	mBuO-	HO-
**45**	12-Deoxyphorbol-13-[8′-oxo- hexadeca-2E,4E,6E-trienoate]	2.2 ± 1.5	5.9	H-	OxoHDTO-	HO-
**46**	12-Deoxyphorbol-13-hexadecanoate	0.02 ± 0.001	1500	H-	HexadecO-	HO-
**47**	12-Deoxy-5β-hydroxy-phorbol-13-hexadecanoate	0.13 ± 0.03	98	H-	HexadecO-	HO-
**48**	12-Deoxy-6,7-epoxy-5β-hydroxy-phorbol-13-hexadecanoate	0.09 ± 0.05	54	-	-	-
**49**	12-Deoxy-5β,6β,7α-trihydroxy-phorbol-13-hexadecanoate	2.14 ± 0.3	26	-	-	-
	**4,12-Dideoxyphorbol esters**					
**50**	4α-12-Dideoxyphorbol-13(2,3-dimethyl)butyrate-20-acetate	>11	n.d	-	dBuO-	OAc-
**51**	4β-12-Dideoxyphorbol-13(2,3-dimethyl)butyrate-20-acetate	4.0 ± 0.3	10.6	-	dBuO-	OAc-
	**Ingenanes**					
**52**	Ingenol	30.1 ± 19.2	4.8	HO-	HO-	-
**53**	Ingenol-3-mebutate	22.9 ± 5.2	2.3	MbO-	HO-	-
**54**	Ingenol-3,20-dibenzoate	1.2 ± 0.1	6.4	BzO-	BzO-	-
	Chloroquine	10 ± 5	8.9	-	-	-

EC50s are given in μM, unless otherwise stated. Values are the median ± median absolute deviation calculated from at least three independent assays. SI, selectivity index, calculated as CC_50_ Vero/EC50 CHIKV. n.d. = not determined (EC50 50% effective concentration or concentration which is calculated to inhibit virus induced cell death by 50%, and CC_50_ 50% antimetabolic concentration or concentration which is calculated to inhibit the overall cell metabolism by 50%). Anti-CHIKV results obtained with the methodology from Bourjot et al. [21].

**Table 2 molecules-24-02336-t002:** Anti-CHIKV activities of daphnanes **55**–**66**.

	Compound	CHIKV EC50	SI
	**Daphnanes**		
**55**	Resiniferatoxin	1.8 ± 0.2	2.3
**56**	Trigocherrierin A	0.6 ± 0.1	71.7
**57**	Trigocherrin A	1.5 ± 0.6	23
**58**	Trigocherrin B	2.6 ± 0.7	36
**59**	Trigocherrin F	3.0 ± 1.2	7.7
**60**	Trigocherriolide A	1.9 ± 0.6	2.4
**61**	Trigocherriolide B	2.5 ± 0.3	2.1
**62**	Trigocherriolide C	3.9 ± 1.0	2.7
**63**	Trigocherriolide E	0.7 ± 0.1	9.4
**64**	Neoguillauminin A	17.7 ± 0.8	2
**65**	Codiapeltine A	10.0 ± 2.3	5
**66**	Codiapeltine B	4.4 ± 0.5	11
	Chloroquine	10 ± 5	8.9

EC50s are given in μM; values are the median ± standard deviation calculated from at least three independent assays. SI, selectivity index, calculated as CC50 Vero/EC50 CHIKV. n.d. = not determined. Anti-CHIKV results obtained with the methodology from Bourjot et al. [21].

**Table 3 molecules-24-02336-t003:** Anti-CHIKV activities of jatrophanes **67**–**92**.

	Compound Name	CHIKV EC50	SI	R1	R2	R3	R4	R5
	**9,14-Dioxojatropha-dienes**							
**67**	3,5,7,8,15-Pentaacetoxy-2-hydroxy-9,14-dioxojatropha-6(17),11E-diene	>164	n.d.	AcO-	H	-	-	-
**68**	3,5,7,15-Tetraacetoxy-2-hydroxy-8-isobutyryloxy-9,14-dioxojatropha-6(17),11E-diene	>196	n.d.	iBuO-	H	-	-	-
**69**	3,5,7,15-Tetraacetoxy-2-hydroxy-8-tigloyloxy-9,14-dioxojatropha-6(17),11E-diene	0.76 ± 0.14	208	TiglO-	H	-	-	-
**70**	3,5,7,15-Tetraacetoxy-8-benzoyloxy-2-hydroxy-9,14-dioxojatropha-6(17),11E-diene	4.3 ± 0.2	29	BzO-	H	-	-	-
**71**	esulatin B	60 ± 14	>2.6	AcO-	AcO-	-	-	-
**72**	2,3,5,7,15- Pentaacetoxy-8-tigloyloxy-9,14-dioxojatropha-6(17),-11E-diene	17.4 ± 0.7	8.3	TiglO-	AcO-	-	-	-
**73**	2,3,5,8,15- Pentaacetoxy-7-benzoyloxy-9,14-dioxojatropha-6(17),11E-diene	17.1	>2.9	BzO-	AcO-	-	-	-
	**9-Oxojatropha-dienes**							
**74**	5,7,14- Triacetoxy-3-benzoyloxy-8,15-dihydroxy-9-oxojat-opha-6(17),11E-diene	19.5 ± 3.6	7.8	AcO-	H	-	-	-
**75**	5,7-Diacetoxy-3-benzoyloxy-14,15-dihydroxy-8-isobutyryloxy-9-oxojatropha-6(17),11E-diene	21.0 ± 3.4	2.8	iBuO-	H	-	-	-
**76**	5,7-Diacetoxy-3-benzoyloxy-14,15-dihydroxy-8-(2-methylbutyryloxy)-9-oxojatropha-6(17),11E-diene	111 ± 14	>1.7	mBuO-	H	-	-	-
**77**	5,7,14-Tri- acetoxy-3-benzoyloxy-15-hydroxy-9-oxojatropha-6(17),11E-diene	80 ± 6	1.9	H	AcO-	-	-	-
	**14** **-Oxojatropha-dienes**							
**78**	Euphodendroidin E	>29.2	n.d	H	AcO-	iBuO-	BzO-	AcO-
**79**	Euphodendroidin F	57.3	1.9	HO-	AcO-	iBuO-	BzO-	AcO-
**80**	Euphodendroidin J	>144.4	n.d.	HO-	BzO-	HO-	BzO-	AcO-
**81**	Euphodendroidin A	>28.6	n.d.	AcO-	H	iBuO-	BzO-	AcO-
**82**	Euphodendroidin K	>124.4	<1.0	AcO-	iBuO-	iBuO-	BzO-	AcO-
**83**	Euphodendroidin L	>44.9	n.d.	AcO-	AcO-	iBuO-	BzO-	AcO-
**84**	Euphodendroidin M	>42.8	n.d.	AcO-	AcO-	iBuO-	iBuO-	AcO-
**85**	Euphodendroidin B	133.6	0.5	AcO-	H	mBuO-	BzO-	AcO-
**86**	Euphodendroidin N	>42.5	1.1	AcO-	H	BzO-	BzO-	AcO-
**87**	Euphodendroidin O	27.4	1.3	AcO-	H	BzO-	BzO-	H
**88**	2,3,5,7,8,9,15-Heptahydroxyjatropha-6(17),11-diene-14-one 2,5,8, 9-tetraacetate-3-(benzoyloxyacetate)-7-(2-methyl-propionate)	5.5 ± 1.7	3.2	AcO-	BzOAcO-	AcO-	iBuO-	AcO-
	**Terracinolides**							
**89**	13α-Terracinolide G	>132.6	n.d.	AcO-	AcO-	HO-	H	-
**90**	13α-Terracinolide B	>125.6	n.d.	AcO-	AcO-	HO-	AcO-	-
**91**	Terracinolide C	15.0 ± 3.8	2.4	AcO-	iBuO-	H	H	-
**92**	Terracinolide J	>135.4	n.d.	H	AcO-	H	AcO-	-
	Chloroquine	10 ± 5	8.9	-	-	-	-	-

EC50s are given in μM, unless otherwise stated. Values are the median ± median absolute deviation calculated from at least three independent assays. SI, selectivity index, calculated as CC50 Vero/EC50 CHIKV. n.d. = not determined. Anti-CHIKV results obtained with the methodology from Bourjot et al. [21].

**Table 4 molecules-24-02336-t004:** Anti-CHIKV activities of premysinanes **93**–**99** and myrsinane **100**.

	Compound	CHIKV EC50	SI	R1	R2	R3	R4	R5
	**Premyrsinol esters**							
**93**	3β,7β,13β,17-*O*-Tetraacetyl-5α-*O*-benzoyl-14-oxopremyrsinol	78	2.2	AcO-	BzO-	AcO-	AcO-	H
**94**	3β,7β,15β,17-*O*-Tetraacetyl-5α-*O*-benzoyl-14-oxopremyrsinol	>152	n.d.	AcO-	BzO-	AcO-	H	AcO-
**95**	3β,7β,13β,17-*O*-Tetraacetyl-5α-*O*-(2-methylbutyryl)-14-oxopre- myrsinol	>50	<5	AcO-	mBuO-	AcO-	AcO-	H
**96**	7β,13β,17-*O*-Triacetyl-5α-*O*-(2-methylbutyryl)-3β-*O*-propanoyl- 14-oxopremyrsinol	107	2.2	PrO-	mBuO-	AcO-	AcO-	H
**97**	7β,17-*O*-Diacetyl-5α-*O*-benzoyl-13β-nicotinyl-3β-*O*-propanoyl- 14-oxopremyrsinol	>107	<2.4	PrO-	BzO-	AcO-	NicO-	H
**98**	13β,17-*O*-Diacetyl-5α-*O*-benzoyl-7β-hydroxy-3β-*O*-propanoyl- 14-oxopremyrsinol	11 ± 1.4	5.8	PrO-	BzO-	H	H	AcO-
**99**	Premyrsinol-3-propanoate-5-benzoate-7,13,17-triacetate	>144	n.d.	PrO-	BzO-	AcO-	AcO-	H
	**Myrsinol ester**							
**100**	5α,7β-*O*-Diacetyl-14β-*O*-benzoyl-3β-*O*-propanoylmyrsinol	84	1.9	-	-	-	-	-
	Chloroquine	10 ± 5	8.9	-	-	-	-	-

EC50s are given in μM, unless otherwise stated. Values are the median ± median absolute deviation calculated from at least three independent assays. SI, selectivity index, calculated as CC50 Vero/EC50 CHIKV. n.d. = not determined. Anti-CHIKV results obtained with the methodology from Bourjot et al. [21].

**Table 5 molecules-24-02336-t005:** Anti-CHIKV activities of flexibilanes **101**–**110**.

	Compound	CHIKV EC50	SI
	**Flexibilanes**		
**101**	tonantzitlolone A	>215	n.d.
**102**	tonantzitlolone B	12 ± 3	10.2
**103**	tonantzitlolone C	24 ± 1	>9
**104**	tonantzitlolone D	>222	n.d.
**105**	tonantzitlolone E	>107	n.d.
**106**	tonantzitlolone F	19 ± 2	3
**107**	tonantzitlolone G	168	>1.3
**108**	tonantzitlolone H	>191	n.d.
**109**	tonantzitlolone I	>208	n.d.
**110**	tonantzitloic acid	>201	n.d.
	chloroquine	10 ± 5	8.9

EC50s are given in μM; values are the median ± standard deviation calculated from at least three independent assays. SI, selectivity index, calculated as CC50 Vero/EC50 CHIKV. n.d. = not determined. Anti-CHIKV results obtained with the methodology from Bourjot et al. [21].

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
