# Peer review of "Macrocyclic Diterpenoids from Euphorbiaceae as A Source of Potent and Selective Inhibitors of Chikungunya Virus Replication"

_molecules, 2019, doi:10.3390/molecules24122336_

Round 1

Reviewer 1 Report

The authors describe diterpenoids from Euphorbiaceae, and their ability to inhibit viral replication. The structure-activity relationships and the possible mechanism of their action are also detailed. This paper is interesting. The referee would agree with the publication of this paper as the review in Molecules. But, please make consideration for minor revision of the following point.

1.       Line 28-41: Authors should make a figure, and put it on the manuscript.

2.       Line 92-98: Authors should cite the figure 2.

3.       Line 141: Authors should describe the explanation of what CC50 Vero and EC50 CHIKV mean.

4.       Table 1-5: Authors should add columns of references in Tables.

Author Response

Reviewer 1

The authors describe diterpenoids from Euphorbiaceae, and their ability to inhibit viral replication. The structure-activity relationships and the possible mechanism of their action are also detailed. This paper is interesting. The referee would agree with the publication of this paper as the review in Molecules. But, please make consideration for minor revision of the following point.

1.       Line 28-41: Authors should make a figure, and put it on the manuscript.

Thank you for your comment. We had indeed thought to illustrate the introduction with one or two figures but we renounced it because we think they would duplicate the figures of the quoted articles (Chem Rev 2008 (3), 2014 (5) and 2015 (4) and ref Appendino (7)).

2.       Line 92-98: Authors should cite the figure 2.

“Figure 2” has been added in the text in line 93.

3.       Line 141: Authors should describe the explanation of what CC50 Vero and EC50 CHIKV mean.

CC50 and EC50 have been explained in caption of Table 1

4.       Table 1-5: Authors should add columns of references in Tables.

Unless stated by editor, we do not wish to add any additional columns to Tables 1-4 due to lack of space. This would require us to reduce the font size or significantly increase the size of the tables.

All correspondences between molecules and references are already indicated at the beginning of each section.

Reviewer 2 Report

Remy and Litaudon wrote a comprehensive  review regarding macrocyclic diterpenoids and their activity against chikungunya virus replication. The most relevant structural requirements necessary for the antiviral biological activity of this important class of natural compounds as well as the major issues regarding their toxicity are clearly and comprehensively described. The review is well organized and could be of interest for a broad readership 

I would suggest the authors to add a figure describing the binding mode of representative diterpenoids to PKC and the related pharmacophoric elements identified.

Minor points:

Figure 2 is never mentioned in the main text. Reference to both Figure 2 and Table 1 should be added in the main text at page 2, lines 92-98

Author Response

Reviewer 2

Remy and Litaudon wrote a comprehensive review regarding macrocyclic diterpenoids and their activity against chikungunya virus replication. The most relevant structural requirements necessary for the antiviral biological activity of this important class of natural compounds as well as the major issues regarding their toxicity are clearly and comprehensively described. The review is well organized and could be of interest for a broad readership 

I would suggest the authors to add a figure describing the binding mode of representative diterpenoids to PKC and the related pharmacophoric elements identified.

Thank you for your comment. The binding mode of TPA and related diterpenoids and the structural determinants critical for the activation of Protein kinase C were analyzed in details by two research groups in 1986 (Wender et al PNAS 1986, 83, 4214-4218 and Jeffrey and Liskamp, PNAS, 1986, 83, 241-245). We discussed in details these aspects and both papers are listed in the reference section. However, we believe that a figure would not bring new elements compared to what was described. 

We add the following sentence line 262: “By using computer-assisted modeling, it has been shown that the pharmacophore model for PEs required a hydrophobic region consisting of acyl substituents on C-12 and/or C-13 and a cyclohexane- cyclopropane-annellated ring system, and a hydrophilic domain spanning the C-3 to C-9 region including four groups able to form hydrogen bonds. An adequate distance and orientation of the ring system relative to membrane lipid bilayer are also necessary [51,65].”

Minor points:

Figure 2 is never mentioned in the main text. Reference to both Figure 2 and Table 1 should be added in the main text at page 2, lines 92-98

This has been corrected.

Reviewer 3 Report

The review by Remy and Litaudon reports on the anti-chikungunya (and anti-HIV) activity of different classes of macrocyclic diterpenoids derived from Euphorbiaceae family plants. The review is well written and organized. The information are reported in a clear and concise manner. Clear SAR insights are delineated within the different classes of diterpenoids.

In my opinion the review is suitable for publication in Molecules.

Minor points:

- Although the authors reported for each of the compounds the selectivity index, they never discuss about the cytotoxicity of the compounds. Perhaps, the authors could add a comment.

 - Section 6 line 247, In reporting the involvement of PKC in the antiviral activity of phorbol esters, the authors should indicate how the information on the interaction between phorbols and PKC has been acquired.

- line 128, the EC50 for compound 32 is reported to be 1.5 μM while in Table 1, it is 1.4 μM.

Author Response

The review by Remy and Litaudon reports on the anti-chikungunya (and anti-HIV) activity of different classes of macrocyclic diterpenoids derived from Euphorbiaceae family plants. The review is well written and organized. The information are reported in a clear and concise manner. Clear SAR insights are delineated within the different classes of diterpenoids.

In my opinion the review is suitable for publication in Molecules.

Minor points:

- Although the authors reported for each of the compounds the selectivity index, they never discuss about the cytotoxicity of the compounds. Perhaps, the authors could add a comment.

Thank you for this comment. We started the discussion of section 1 by the following sentences:First the cytotoxicity of all compounds was evaluated against African green monkey kidney epithelial cell line (Vero cells). The CC50 values ranged from 4.1 to > 343 µM, phorbol (1) being the less cytotoxic compound. Among compounds with a selective index > 20 (vide infra), the highest cytotoxicity was obtained for compounds with a long acyl chain either at C-12 or C-13 position (11, 15 and 48).”

 - Section 6 line 247, In reporting the involvement of PKC in the antiviral activity of phorbol esters, the authors should indicate how the information on the interaction between phorbols and PKC has been acquired.

Thank you for your comment. We slightly modified the text in line 257 and indicated how information on the interaction between phorbols and PKC has been acquired.

The sentence in line 262 is: “By using computer-assisted modeling, it has been shown that the pharmacophore model for PEs required a hydrophobic region consisting of acyl substituents on C-12 and/or C-13 and a cyclohexane- cyclopropane-annellated ring system, and a hydrophilic domain spanning the C-3 to C-9 region including four groups able to form hydrogen bonds. An adequate distance and orientation of the ring system relative to membrane lipid bilayer are also necessary [51,65].”

- line 128, the EC50 for compound 32 is reported to be 1.5 μM while in Table 1, it is 1.4 μM.

This has been corrected.